# MaskMed: Decoupled Mask and Class Prediction for Medical Image Segmentation

## Abstract

Medical image segmentation typically adopts a point-wise convolutional segmentation head to predict dense labels, where each output channel is heuristically tied to a specific class. This rigid design limits both feature sharing and semantic generalization. In this work, we propose a unified decoupled segmentation head that separates multi-class prediction into class-agnostic mask prediction and class label prediction using shared object queries. Furthermore, we introduce a Full-Scale Aware Deformable Transformer module that enables low-resolution encoder features to attend across full-resolution encoder features via deformable attention, achieving memory-efficient and spatially aligned full-scale fusion. Our proposed method, named MaskMed, achieves state-of-the-art performance, surpassing nnUNet by +2.0% Dice on AMOS 2022 and +6.9% Dice on BTCV.

## 1 Introduction

Medical image segmentation plays a critical role in a wide range of clinical and diagnostic applications, including organ delineation, tumor quantification, and treatment planning. The dominant paradigm in modern medical segmentation systems, such as UNet Ronneberger et al. (2015) and its variants (e.g., nnUNet Isensee et al. (2021), UNETR Hatamizadeh et al. (2022), nnFormer Zhou et al. (2021), SwinUNETR Tang et al. (2022)), adopts a simple yet effective architecture: an encoder–decoder architecture coupled with a lightweight segmentation head. In most cases, this segmentation head consists of a point-wise convolution that directly maps deep feature channels to class logits, followed by a softmax activation and optimized using cross-entropy or Dice loss.

Despite its empirical success, the widely adopted point-wise convolutional segmentation head introduces a rigid inductive bias in how semantic classes are predicted. Specifically, it can be interpreted as a learnable linear projection matrix multiplied by an explicitly predefined channel-to-class identity matrix. The learnable linear projection matrix maps high-level feature representations to a fixed number of output channels, while the manual channel-to-class identity matrix enforces a one-to-one mapping between channel indices and semantic category labels. This hand-crafted structure assumes that each output channel is solely responsible for a specific class, regardless of context.

Such a design imposes two key limitations. First, it restricts the model's ability to learn flexible, data-driven associations between spatial patterns and class semantics. The head operates independently per channel, preventing joint reasoning across categories. Second, it discards the potential for feature sharing or contextual adaptation, which are essential for modeling ambiguous or overlapping structures in medical images.

To address these limitations, we propose a unified and fully decoupled segmentation head that predicts sets of binary masks and their associated class labels in parallel, without hard-coded bindings between channels and classes. This design enables dynamic, compositional reasoning, where mask and class embeddings are learned independently and can interact adaptively. Notably, traditional point-wise convolutional heads can be viewed as a special case of our framework, where class embeddings are fixed as identity mappings and masks are generated by direct channel-wise linear projection. By breaking this rigid structure, our unified segmentation head improves both the expressiveness and performance of medical image segmentation models.

Another insight is that, although spatial granularity varies across decoder stages, the semantic queries responsible for mask and class prediction should remain consistent. Therefore, we introduce a shared

query mechanism across all decoder levels, where queries and masks are propagated hierarchically and refined via masked cross-attention. This mechanism allows low-level predictions to guide higher-level attention toward important spatial regions, leading to improved segmentation accuracy. This design forms our final *Masked Multi-Scale Segmentation Head* architecture, which effectively integrates multi-scale information while maintaining semantic coherence across decoder stages.

To further enhance the representational capacity and information flow between the encoder and decoder, we explore more effective strategies for fusing multi-scale features. A straightforward approach is to apply attention-based modules to the encoder's hierarchical outputs for global context modeling. However, such designs are typically memory-intensive and limited to low-resolution features due to the quadratic complexity of standard attention mechanisms. Inspired by deformable attention Zhu et al. (2021); Cheng et al. (2022), we propose the *Full-Scale Aware Deformable Transformer (FSAD-Transformer)*, which enables efficient and adaptive aggregation of features across all resolution levels of the encoder with low computational cost.

Our main contributions can be summarized as follows:

- We propose MaskMed, a unified segmentation head framework that generalizes traditional point-wise segmentation heads into a fully decoupled design, where binary mask prediction and class prediction are independently learned and jointly optimized.
- To establish precise supervision, we introduce bipartite matching between predicted mask-label pairs and ground truth, marking the first application of this strategy in medical image segmentation.
- We develop a *Masked Multi-Scale Segmentation Head* that uses a shared query mechanism across decoder stages, enabling consistent semantic prediction while refining spatial details hierarchically with masked attention.
- We introduce the *Full-Scale Aware Deformable Transformer* to efficiently fuse full-scale encoder features by attending to sparse yet informative regions across all spatial resolutions.
- MaskMed achieves new state-of-the-art performance, outperforming nnUNet by +2.0% Dice on AMOS 2022 Ji et al. (2022) and +6.9% Dice on BTCV Landman et al. (2015).

## 2 RELATED WORK

**Medical Image Segmentation.** Deep learning has significantly advanced medical image segmentation, with fully convolutional architectures like U-Net Ronneberger et al. (2015) serving as the foundation. nnU-Net Isensee et al. (2021) further automated architecture and hyperparameter tuning, achieving state-of-the-art performance across numerous 3D benchmarks. With the emergence of transformers, a new wave of methods has extended these architectures by incorporating self-attention mechanisms, including UNETR Hatamizadeh et al. (2022), Swin UNETR Tang & et al. (2022), and nnFormer Zhou et al. (2021). These models typically adopt an encoder–decoder framework where transformers enhance long-range dependency modeling. However, despite innovations in feature encoding, most existing methods still rely on *simple point-wise convolution segmentation heads* for final mask prediction. These segmentation heads operate by projecting the high-dimensional feature maps into a fixed number of output channels, where each channel is assigned to a specific semantic class. This rigid channel-to-class assignment enforces a hard-coded one-to-one mapping, which poses two major limitations: i) it constrains the model's expressiveness by preventing dynamic or compositional reasoning across classes, and ii) it treats all output channels independently, ignoring potential inter-class correlations or shared structures. Such designs lack flexibility and scalability.

**Mask Classification and Decoupled Prediction.** In the vision community, recent work such as MaskFormer Cheng & et al. (2021) and Mask2Former Cheng & et al. (2022) proposed a new paradigm for segmentation: treating it as a *mask classification task*. Instead of directly predicting multi-class maps, these approaches decouple the task into predicting a set of class-agnostic binary masks and their associated class labels, enabling better generalization and alignment with instance-level structures. This formulation also naturally facilitates the use of bipartite matching (e.g., via the Hungarian algorithm) for optimal supervision. In medical image segmentation, this decoupled formulation has not been explored. To the best of our knowledge, we are the first to adopt a fully decoupled design with bipartite matching, enabling more flexible and instance-aware segmentation.

**Deformable Modules in Vision** Deformable modules have significantly advanced visual recognition by adapting receptive fields to object shapes and scales, such as Deformable Convolutional Networks

Figure 1: Illustration of different segmentation head architectures evolving from the conventional UNet-based design (a, b) to our proposed decoupled mask and class embedding framework (f).

(DCNs) Dai et al. (2017), introduced learnable spatial offsets into convolutional kernels, enabling spatially adaptive feature extraction. This idea was later extended to the attention paradigm. In particular, Deformable DETR Zhu et al. (2021) and Xia et al. (2022) proposed a sparse, content-adaptive attention mechanism that restricts computation to a small set of key sampling points, dramatically reducing the memory and convergence burden of standard transformers. In medical imaging, deformable components have been used for tasks like registration Chen et al. (2022), but their use in segmentation remains limited. We address this gap by introducing a Full-Scale Aware Deformable Transformer that applies deformable attention for efficient, global feature fusion across encoder scales in high-resolution 3D medical segmentation.

## 3 THE PROPOSED METHOD

### 3.1 RETHINKING SEGMENTATION HEADS.

Medical image segmentation is conventionally formulated using convolution-based encoder–decoder architectures such as U-Net, as illustrated in Fig.1(a). These models typically attach multiple point-wise convolutional segmentation heads (SegHeads) at different decoding stages (Fig.1(b)). The encoder extracts hierarchical feature representations, while the decoder reconstructs a high-resolution feature map $\mathbf{F} \in \mathbb{R}^{B \times C \times D \times H \times W}$ enriched with high-level semantic information, where $B$ is the batch size, $C$ denotes the number of feature channels, and $(D, H, W)$ are the spatial dimensions. Each SegHead, typically implemented as a point-wise convolution, projects the feature map from $C$ channels to $N$ class logits, yielding an output of shape $(B, N, D, H, W)$, where $N$ denotes the number of semantic classes. A softmax operation is applied along the channel dimension to yield voxel-wise class probabilities.

As illustrated in Fig. 1(b), the point-wise convolutional SegHead can be interpreted as a learnable linear projection with parameterized by a weight matrix of size $C \times N$. This layer maps the semantically rich decoder features to a set of logit masks, each corresponding to a specific semantic class. Conceptually, this is followed by an implicit identity matrix $\mathbf{I}_{cls} \in \mathbb{R}^{N \times N}$, which enforces a rigid one-to-one assignment between output channels and semantic classes. Under this formulation, each channel index in the logit masks is assumed to represent a single semantic class exclusively (*i.e.,* the $i$-th channel contributes 100% to the $i$-th class). This fixed channel-to-class mapping is consistent with the one-hot assumption embedded in the identity matrix. The final predictions are obtained by applying a softmax operation along the channel dimension, and supervised using a combination of Dice loss and Binary Cross Entropy (BCE) loss against the ground truth labels. The formulation:

$$\mathcal{L} = \sum_{i=1}^{N} \mathcal{L}_{\text{Dice+BCE}} \left( \hat{\mathbf{Y}}_i, \mathbf{Y}_i \right), \quad \hat{\mathbf{Y}}_i = \text{softmax}([\mathbf{F} \cdot \mathbf{M}_{\text{mask}} \cdot \mathbf{I}_{\text{cls}}]_i) \tag{1}$$

To enable more flexible and expressive representation learning, we progressively evolve this baseline through a series of architectural modifications:

- **Learnable Class Embedding via Point-wise Projection (Fig. 1(c))**: To move beyond the rigid one-hot assumption imposed by the identity matrix, we replace the fixed class embeddings with a learnable parameter matrix implemented as a point-wise $(1 \times 1 \times 1)$ convolution. This modification enables the model to learn the contribution of each logit mask to the final class prediction in a flexible manner, rather than relying on static, predefined mappings. Each class embedding becomes

a trainable vector that can better capture inter-class relationships and adapt to the feature distribution during training. The formulation can be expressed as:

$$\mathcal{L} = \sum_{i=1}^{N} \mathcal{L}_{\text{Dice+BCE}} \left( \hat{\mathbf{Y}}_i, \mathbf{Y}_i \right), \quad \hat{\mathbf{Y}}_i = \text{softmax}([\mathbf{F} \cdot \mathbf{M}_{\text{mask}} \cdot \mathbf{M}_{\text{cls}}]_i) \tag{2}$$

- **Data-Driven Class Embedding via Transformer (Fig. 1(d))**: To further enhance the modeling capacity and contextual awareness of the class embeddings, we incorporate a Transformer-based module consisting of self-attention, cross-attention, and feed-forward layers. Initialized with a set of learnable queries $\mathbf{E}_{\text{Q}} \in \mathbb{R}^{N \times E}$, this module dynamically generates class embeddings by conditioning on the semantic feature maps. This fully data-driven formulation allows the class embeddings to be informed by global context and spatial semantics, providing a more expressive alternative to static or point-wise embeddings. The formulation can be expressed as:

$$\mathcal{L} = \sum_{i=1}^{N} \mathcal{L}_{\text{Dice+BCE}} \left( \hat{\mathbf{Y}}_i, \mathbf{Y}_i \right), \quad \hat{\mathbf{Y}}_i = \text{softmax}([\mathbf{F} \cdot \mathbf{M}_{\text{mask}} \cdot f(\mathbf{E}_{\text{Q}}, \mathbf{F})_{\text{cls}}]_i) \tag{3}$$

- **Data-Driven Mask and Class Embedding via Transformer (Fig. 1(e))**: We further enhance the segmentation head by replacing the point-wise convolutional mask embedding with a Transformer-based module, mirroring the design of the class embedding branch. This design enables both class and mask embeddings to be learned in a unified, data-driven manner. The formulation can be expressed as:

$$\mathcal{L} = \sum_{i=1}^{N} \mathcal{L}_{\text{Dice+BCE}} \left( \hat{\mathbf{Y}}_i, \mathbf{Y}_i \right), \quad \hat{\mathbf{Y}}_i = \text{softmax}([\mathbf{F} \cdot f(\mathbf{E}_{\text{Q}}, \mathbf{F})_{\text{mask}} \cdot f(\mathbf{E}_{\text{Q}}, \mathbf{F})_{\text{cls}}]_i) \tag{4}$$

- **Decoupled Multi-Class Masks to Binary Masks with Classifiers (Fig. 1(f))**: We propose a fully decoupled segmentation head that separates multi-class mask prediction into binary mask prediction with class prediction. Specifically, the model predicts $N$ class-agnostic binary masks independently, where each mask captures a potential object or region of interest. These masks are supervised using standard binary segmentation losses (e.g., Dice + BCE). In parallel, a classification branch assigns a semantic class to each predicted binary mask by computing a class token and applying a cross-entropy loss. To replace the rigid channel-to-class mapping in conventional segmentation heads, we introduce a bipartite matching mechanism during training. This enables each predicted mask-class pair (i.e., object query) to flexibly match the ground truth with the best similarity. Given a matching $\sigma$, the formulation can be expressed as:

$$\mathcal{L} = \sum_{i=1}^{N} \left[ \mathcal{L}_{\text{Dice+BCE}} \left( \text{sigmoid} \left( [\mathbf{F} \cdot f(\mathbf{E}_{\text{Q}}, \mathbf{F})_{\text{mask}}]_{\sigma(i)} \right), \mathbf{Y}_i \right) + \mathcal{L}_{\text{CE}} \left( [f(\mathbf{E}_{\text{Q}}, \mathbf{F})_{\text{cls}}]_{\sigma(i)}, \mathbf{y}_i \right) \right] \tag{5}$$

Notably, all previous segmentation head variants, ranging from the classical UNet with fixed identity-based class embeddings to transformer-enhanced designs (Fig.1(c–e)), can be regarded as special cases of our final decoupled formulation (Fig.1(f)). These earlier designs implicitly entangle mask prediction and classification via fixed projections or shared embeddings. In contrast, our fully decoupled design explicitly separates the generation of binary masks from semantic classification, allowing each to be optimized independently. This separation not only removes rigid architectural assumptions (e.g., one-hot class encodings) but also improves modeling flexibility, facilitates overlapping or ambiguous regions, and enhances generalization to complex scenarios in medical image segmentation.

## 3.2 MASKED MULTI-SCALE SEGMENTATION HEAD.

Building upon the unified segmentation head introduced in the previous section, we propose a series of architectural refinements aimed at improving both efficiency and segmentation quality. In conventional UNet-style architectures, deep supervision is commonly employed, where each decoder stage produces predictions at different resolutions to accelerate convergence. We adopt a similar approach, applying deep supervision by assigning an independent query set to each decoder stage.

However, this setup neglects semantic consistency across stages. While spatial resolution varies, the underlying semantic queries responsible for mask and class prediction should ideally remain coherent.

Figure 2: **Model Architecture Overview.** (a) Our full model adopts an encoder-decoder framework with a Full-Scale Aware Deformable Transformer (FSAD-Transformer) module bridging multi-scale encoder features and decoder inputs. (b) The Masked Multi-Scale Segmentation Head uses a shared query set to decode both mask and class embeddings via transformer layers. (c) The FSAD-Transformer allows deformable attention across the full feature hierarchy, using multi-scale queries and full-resolution value features.

To address this, we introduce a shared query mechanism across all decoder stages. A single set of learnable queries is initialized at the lowest-resolution decoder output, and the output from each stage's transformer is propagated upward as input to the next higher-resolution stage. This design enforces cross-scale consistency, encourages more stable optimization, and enhances generalization.

To further improve stage-wise feature refinement, we introduce a masked attention mechanism. Specifically, the predicted masks from a lower-resolution stage are used as spatial priors for the next stage's transformer, effectively guiding attention toward task-relevant regions and suppressing background noise. This leads to our proposed Masked Multi-Scale Segmentation Head, illustrated in Fig. 1(a), (b).

Fig. 1(b) also details the inner structure of our transformer module. Each transformer block receives a shared set of learnable queries as input. At each stage, the decoder feature map $\mathbf{F}$ serves as the key and value in the masked cross-attention. To reduce memory consumption, especially for high-resolution features, we apply spatial average pooling on $\mathbf{F}$ to reduce its spatial dimensions before use. The transformer block includes masked cross-attention, followed by standard self-attention and feed-forward layers. The output hidden features are passed to the next stage as the input query. Meanwhile, two lightweight MLP branches are employed to predict the mask embeddings and class embeddings. The mask embedding is multiplied with the input feature map $\mathbf{F}$ to obtain dense segmentation masks. The class embedding is used to predict the associated semantic classifier.

### 3.3 FULL-SCALE AWARE DEFORMABLE TRANSFORMER

Building upon the previous section, which established an encoder-decoder structure enhanced by a unified Masked Multi-Scale Segmentation Head, we next focus on improving the fusion of encoder features into the decoder via skip connections. This component is essential for preserving spatial details and achieving accurate high-resolution segmentation.

A straightforward strategy would be to enhance the skip-connected encoder features using a standard Transformer, capturing long-range dependencies across the spatial dimensions. Unlike the object query to image feature attention paradigm used in the segmentation head, this module requires attention among the image features themselves, capturing intra-feature context, which results in huge memory consumption. Due to memory constraints inherent to volumetric medical data, a standard Transformer can only process a limited number of scales of skip connections. In our initial exploration, we applied Transformers only to the bottom three encoder stages, which have the smallest spatial resolutions. Higher-resolution features had to be excluded from global reasoning, resulting in suboptimal feature integration.

To address this, we explored replacing self-attention with deformable attention, inspired by Deformable DETR. By focusing computation on a sparse set of sampling points, deformable attention dramatically reduces memory usage while retaining the ability to learn semantically meaningful receptive fields. This allowed us to incorporate four encoder stages into the global context modeling. Nevertheless, higher-resolution encoder features still remained unused, limiting the effectiveness of full-scale fusion.

Our key insight is that, although feature maps from different stages vary in resolution, they encode spatially aligned structures. Voxels at corresponding positions across scales describe the same

| Method | Spleen | R.Kd | L.Kd | GB | Eso. | Liver | Stom. | Aorta | IVC | Panc. | RAG | LAG | Duo. | Blad. | Pros. | Average |
|---|---|---|---|---|---|---|---|---|---|---|---|---|---|---|---|---|
| UNETR Hatamizadeh et al. (2022) | 0.928 | 0.913 | 0.903 | 0.719 | 0.763 | 0.955 | 0.849 | 0.922 | 0.838 | 0.766 | 0.663 | 0.663 | 0.662 | 0.815 | 0.744 | 0.807 |
| nnFormer Zhou et al. (2021) | 0.950 | 0.948 | 0.944 | 0.789 | 0.784 | 0.967 | 0.914 | 0.931 | 0.868 | 0.828 | 0.654 | 0.695 | 0.759 | 0.865 | 0.773 | 0.845 |
| SwinUNETR Hatamizadeh et al. (2021) | 0.954 | 0.954 | 0.950 | 0.819 | 0.852 | 0.972 | 0.919 | 0.955 | 0.911 | 0.875 | 0.775 | 0.801 | 0.816 | 0.895 | 0.812 | 0.884 |
| SwinUNETRv2 He et al. (2023) | 0.959 | 0.962 | 0.958 | 0.842 | 0.867 | 0.976 | 0.933 | 0.957 | 0.920 | 0.889 | 0.783 | 0.812 | 0.843 | 0.913 | 0.836 | 0.897 |
| 3D UX-Net Lee et al. (2022) | 0.955 | 0.956 | 0.953 | 0.826 | 0.858 | 0.972 | 0.922 | 0.955 | 0.915 | 0.881 | 0.781 | 0.809 | 0.820 | 0.902 | 0.823 | 0.889 |
| nn-UNet Isensee et al. (2019) | 0.951 | 0.961 | 0.956 | 0.826 | 0.869 | 0.973 | 0.931 | 0.957 | 0.923 | 0.880 | 0.784 | 0.809 | 0.846 | 0.898 | 0.827 | 0.893 |
| MaskSAM Xie et al. (2024) | 0.963 | **0.973** | **0.969** | 0.872 | 0.876 | **0.982** | 0.940 | **0.962** | 0.922 | 0.888 | 0.794 | 0.813 | 0.851 | 0.920 | 0.854 | 0.905 |
| MaskFormer Cheng & et al. (2021) | 0.946 | 0.959 | 0.953 | 0.790 | 0.835 | 0.969 | 0.914 | 0.948 | 0.905 | 0.853 | 0.735 | 0.671 | 0.814 | 0.865 | 0.787 | 0.863 |
| Mask2Former Cheng & et al. (2022) | 0.954 | 0.966 | 0.962 | 0.819 | 0.855 | 0.972 | 0.934 | 0.954 | 0.917 | 0.874 | 0.760 | 0.779 | 0.844 | 0.891 | 0.809 | 0.886 |
| MaskMed (Ours) | **0.971** | 0.970 | **0.969** | **0.880** | **0.890** | 0.981 | **0.950** | 0.961 | **0.929** | **0.904** | **0.801** | **0.826** | **0.875** | **0.931** | **0.859** | **0.913** |

Table 1: Comparison of MaskMed with state-of-the-art methods on the AMOS test set, evaluated by Dice Score. For a fair comparison, all results are based on 5-fold cross-validation without any ensembles. **Bold** indicates the best. Both MaskFormer and Mask2Former are adapted to 3D.

anatomical region at different semantic levels. To leverage this, we propose the Full-Scale Aware Deformable Transformer (Fig. 1(c)). In this design, we use full-scale encoder features as the value inputs, while queries are extracted only from the lowest four stages. The key innovation lies in enabling each query to attend to full-scale features across all encoder stages, thereby promoting full-resolution context awareness.

To further reduce memory overhead, query-specific attention offsets and weights are learned via compact MLP projections, following the deformable attention paradigm. Each query aggregates information from a sparse set of $K$ sampling points across the full feature pyramid, allowing it to access rich multi-scale context at minimal computational cost. This architecture not only enables dense, deformable, and full-scale feature fusion, but also aligns well with the anatomical consistency across spatial scales in medical imaging.

### 3.4 THE PROPOSED ARCHITECTURE

As illustrated in Fig. 2 (a), our framework, named MaskMed, adopts a hierarchical encoder-decoder design enhanced with a full-resolution cross-scale fusion module and a unified segmentation head. The encoder extracts multi-scale features from volumetric medical images, which are then progressively decoded via upsampling blocks and skip connections. To effectively bridge the semantic gap between encoder and decoder features, we introduce a *Full-Scale Aware Deformable Transformer (FSAD-Transformer)* that enables efficient, deformable attention across the entire feature hierarchy.

At the end of the decoder, we deploy a *Masked Multi-Scale Segmentation Head*, which leverages a shared query set and hierarchical attention refinement to produce accurate mask and class predictions. Unlike traditional segmentation heads that rely on point-wise convolutions and fixed class-channel mappings, our segmentation head produces class-agnostic binary masks and corresponding semantic class embeddings in a decoupled fashion. By avoiding rigid one-hot mappings and incorporating hierarchical masked attention, the head enables consistent and interpretable predictions across scales.

Our proposed unified design integrates full-resolution deformable context aggregation and a unified query-driven segmentation head into a single framework. This enables dense, instance-aware predictions while maintaining spatial precision and semantic consistency. By decoupling binary mask generation from class assignment and enforcing full-scale feature interaction, the model achieves greater flexibility, improved generalization, and stronger interpretability in medical segmentation.

### 3.5 MATCHING AND LOSSES.

Following Cheng et al. (2021); Carion et al. (2020), the overall loss function comprises a standard cross-entropy loss for class predictions and a combination of binary cross-entropy and Dice loss for the final binary mask predictions ($\mathcal{L}_{mask}^{final}$). To determine the optimal one-to-one assignment between predictions and ground truth, we employ bipartite matching Cheng et al. (2021); Carion et al. (2020) between the ground truth segments and the final predictions at the final stage of the segmentation head. This process yields a set of matched indices from the $N$ candidate pairs of binary mask predictions and class predictions. These indices are then used consistently across all segmentation head stages to compute the training losses. Importantly, bipartite matching is performed only once per forward pass, based on the final-stage predictions.

Specifically, the desired final output is represented as $z = \{(p_i, m_i)\}_{i=1}^{N}$, where $N$ pairs of binary masks $\{m_i^{final}|m_i^{final} \in [0,1]^{H \times W}\}_{i=1}^{N}$ are associated with class probability distributions $p_i \in \Delta^{K+1}$. The distribution includes $K$ category labels and an auxiliary "no object" label ($\varnothing$). The set of $N^{gt}$ ground truth segments is represented as $z^{gt} = \{(c_i^{gt}, m_i^{gt})|c_i^{gt} \in \{1,...,K\}, m_i^{gt} \in \{0,1\}^{H \times W}\}_{i=1}^{N^{gt}}$.

| Method | Spl. | R.Kd | L.Kd | GB | Eso. | Liv. | Stom. | Aorta | IVC | Veins | Panc. | AG | DSC |
|---|---|---|---|---|---|---|---|---|---|---|---|---|---|
| TransUNet Chen et al. (2021) | 0.952 | 0.927 | 0.929 | 0.662 | 0.757 | 0.969 | 0.889 | **0.920** | 0.833 | 0.791 | 0.775 | 0.637 | 0.838 |
| 3D UX-Net Lee et al. (2022) | 0.946 | 0.942 | 0.943 | 0.593 | 0.722 | 0.964 | 0.734 | 0.872 | 0.849 | 0.722 | 0.809 | 0.671 | 0.814 |
| UNETR Hatamizadeh et al. (2022) | 0.968 | 0.924 | 0.941 | 0.750 | 0.766 | 0.971 | 0.913 | 0.890 | 0.847 | 0.788 | 0.767 | 0.741 | 0.856 |
| Swin-UNETR Hatamizadeh et al. (2021) | **0.971** | 0.936 | 0.943 | **0.794** | 0.773 | **0.975** | 0.921 | 0.892 | 0.853 | **0.812** | 0.794 | **0.765** | 0.869 |
| nnUNet Isensee et al. (2019) | 0.942 | 0.894 | 0.910 | 0.704 | 0.723 | 0.948 | 0.824 | 0.877 | 0.782 | 0.720 | 0.680 | 0.616 | 0.802 |
| nnFormer Zhou et al. (2021) | 0.935 | 0.949 | 0.950 | 0.641 | **0.795** | 0.968 | 0.901 | 0.897 | 0.859 | 0.778 | 0.856 | 0.739 | 0.856 |
| MaskFormer Cheng & et al. (2021) | 0.963 | 0.946 | 0.950 | 0.561 | 0.755 | 0.969 | 0.892 | 0.894 | 0.858 | 0.738 | 0.826 | 0.674 | 0.835 |
| Mask2Former Cheng & et al. (2022) | 0.965 | 0.949 | 0.950 | 0.626 | 0.772 | 0.970 | 0.897 | 0.897 | 0.865 | 0.752 | 0.835 | 0.706 | 0.849 |
| MaskMed (Ours) | 0.970 | **0.952** | **0.956** | 0.680 | 0.792 | 0.974 | **0.922** | 0.915 | **0.887** | 0.799 | **0.875** | 0.742 | **0.872** |

Table 2: Comparison of MaskMed with state-of-the-art methods on BTCV dataset (DSC in %). The best results are highlighted in **bold**. Both MaskFormer and Mask2Former are adapted to 3D.

Since we set $N \geq N^{gt}$, the ground truth set is padded with "no object" tokens ($\varnothing$) to enable one-to-one matching. Given a matching $\sigma$, the main loss is formulated as:

$$\mathcal{L}_{\text{mask-cls}} = \sum_{l=1}^{L} w_l \cdot \sum_{j=1}^{N} [-\log p_{\sigma(j)}(c_j^{gt}) + \mathbb{1}_{c_j^{gt} \neq \varnothing} \mathcal{L}_{mask}^{\text{final}}(m_{\sigma(j)}^{\text{final}}, m_j^{gt})]. \quad (6)$$

where the $w_i$ denotes the deep supervision weights for different stages. The $w_i$ decrease by half with each reduction in resolution (*i.e.,* $w_2 = \frac{1}{2}w_1$, $w_3 = \frac{1}{4}w_1$). The weights are normalized to sum to 1. Additionally, the resolution of $w_1$ is twice that of $w_2$ and four times that of $w_3$.

## 4 EXPERIMENTS

**Datasets and Evaluation Metrics:** We conduct experiments using two publicly available datasets: the AMOS22 Abdominal CT Organ Segmentation dataset Ji et al. (2022) and the BTCV challenge dataset Landman et al. (2015). **(i)** The AMOS22 dataset contains 300 abdominal CT scans with manual annotations for 16 anatomical structures, which serve as the basis for multi-organ segmentation tasks. The testing set comprises 200 images, and we evaluate our model using the AMOS22 leaderboard. **(ii)** The BTCV dataset includes 30 cases of abdominal CT scans. Following established split strategies Hatamizadeh et al. (2021), we use 24 cases for training and 6 cases for validation. Performance is assessed using average Dice Similarity Coefficient (DSC) across 13 abdominal organs.

### 4.1 COMPARISON WITH STATE-OF-THE-ART METHODS

**Results on AMOS 2022 Dataset.** We evaluate our method, MaskMed, on the AMOS 2022 benchmark and compare it against several recent state-of-the-art 3D medical image segmentation models, including CNN-based (nnU-Net Isensee et al. (2019); Lee et al. (2022)), transformer-based (UN-ETR Hatamizadeh et al. (2022), SwinUNETR Hatamizadeh et al. (2021), and nnFormer Zhou et al. (2021)), and hybrid approaches (3D UX-Net Lee et al. (2022)). As shown in Tab. 1, our method achieves the highest average Dice Score of **0.913**, outperforming the strong baseline nnU-Net (0.893) by a margin of +2.0%, and surpassing the best-performing prior method, MaskSAM (0.905), by +0.8%. To ensure a fair comparison, we adapt both MaskFormer and Mask2Former to the 3D setting by converting all convolutional and attention modules to their 3D counterparts. Our MaskMed outperforms 3D MaskFormer (+5.0%) and 3D Mask2Former (+2.7%) on AMOS.

MaskMed delivers consistent improvements across a wide range of anatomical structures. Notably, it achieves the best Dice on 12 out of 15 organs, including large improvements on challenging regions such as the Esophagus (0.890 *vs.* 0.876 of MaskSAM), Duodenum (0.875 *vs.* 0.851), and Pancreas (0.904 *vs.* 0.889). On major organs like the Right Kidney, Liver, and Aorta, our model also achieves top performance with scores of 0.970, 0.981, and 0.961 respectively. These results highlight the effectiveness of our decoupled segmentation framework and the full-scale deformable fusion design in capturing both fine-grained spatial details and high-level semantic context for complex volumetric segmentation tasks.

**Results on BTCV Dataset.** We present the quantitative results of our experiments on the BTCV dataset in Tab. 2, comparing our proposed MaskMed against several leading convolution-based methods ( nnUNet Isensee et al. (2019)), transformer-based methods (TransUNet Chen et al. (2021), SwinUNet Cao et al. (2021), nnFormer Zhou et al. (2021)). MaskMed achieves the highest average Dice score of **0.872**. For fair comparison, both MaskFormer and Mask2Former are also adapted to 3D. our method MaskMed outperforming 3D MaskFormer (83.5%) and 3D Mask2Former (84.9%) by +3.7% and +2.3%, respectively.

| Method | DSC |
|---|---|
| nnU-Net (fixed CLS Emb) | 0.878 |
| nnU-Net w/ Larger Decoder | 0.890 |
| MLP-based CLS Emb | 0.880 |
| Attention-based CLS Emb | 0.885 |
| Attention-based CLS & Mask Emb | 0.891 |
| Decouple CLS & Mask Emb | 0.903 |

Table 3: Different SegHeads.

| Method | DSC |
|---|---|
| One query per stage | 0.903 |
| One query per model (One_Q / M) | 0.906 |
| One_Q / M + Masked Attn | 0.909 |
| One_Q / M + Masked Attn + StandardTrans | 0.901 |
| One_Q / M + Masked Attn + DeformableTrans | 0.907 |
| One_Q / M + Masked Attn + FSAD-Transformer | 0.913 |

Table 4: Different proposed modules.

| CNN : Transformer | DSC |
|---|---|
| 1:1 | collapse |
| 1:0.5 | 0.822 |
| 1:0.1 | 0.913 |
| 1:0.05 | 0.908 |

Table 5: Learning rate ratios.

| # Object Query | DSC |
|---|---|
| $1 \times N$ | 0.913 |
| $2 \times N$ | 0.904 |
| $3 \times N$ | 0.900 |
| $4 \times N$ | 0.891 |

Table 6: # Object query.

| $\lambda_{\text{Class}} : \lambda_{\text{Mask}_{\text{BCE}}} : \lambda_{\text{Mask}_{\text{Dice}}}$ | DSC |
|---|---|
| $2 : 5 : 5$ | 0.902 |
| $4 : 5 : 5$ | 0.889 |
| $2 : 10 : 10$ | 0.913 |
| $4 : 10 : 10$ | 0.910 |

Table 7: Loss ratios.

Our model delivers state-of-the-art accuracy on 5 out of 13 organs, including R. Kidney (0.952), L. Kidney (0.956), IVC (0.887), and Pancreas (0.875). The consistent gains across both large (e.g., Liver 0.974) and small structures (e.g., Pancreas, Veins) demonstrate the robustness of our decoupled head and deformable feature fusion design.

## 4.2 Ablation Study on Architecture

**Segmentation Head Variants.** As shown in Tab. 3, starting from the baseline nnU-Net with fixed class embeddings (0.878 DSC), replacing the class projection with an MLP brings a small improvement (0.880), while introducing attention-based class embeddings further increases performance to 0.885. Adding attention-based mask embeddings boosts results to 0.891. The best performance is achieved by fully decoupling class and mask embeddings, reaching 0.903 DSC. These results confirm the effectiveness of our decoupled design in enabling more flexible and expressive segmentation.

**Masked Multi-Scale Segmentation Head.** Tab. 4 shows that using a shared query set across decoder stages improves DSC from 0.903 to 0.906. Introducing inter-stage masked attention further increases it to 0.909, demonstrating that spatial priors propagated across scales enhance coherence and prediction quality.

**Full-Scale Aware Deformable Transformer.** Tab. 4 shows that replacing standard skip connections with a Transformer bridge leads to lower performance (0.901) due to limited scale coverage and high memory cost. Switching to deformable attention improves performance to 0.907. Our Full-Scale Aware Deformable Transformer achieves the highest DSC of 0.913 by enabling dense, memory-efficient, and anatomically aligned feature fusion across all encoder stages.

**Impact of Model Capacity.** To ensure the performance gain is not simply due to increased parameters, we scaled up the nnU-Net decoder to match our segmentation head in size in Tab. 3. This led to only a minor DSC improvement from 0.878 to 0.890, while our full model achieves 0.903. The 1.3% gap confirms the effectiveness of our design beyond parameter count.

## 4.3 Ablation on Optimization Sensitivity.

To investigate the optimization dynamics and hyperparameter sensitivity of our proposed model, we conduct ablation studies on three critical factors: (1) learning rate ratios between CNN and Transformer components, (2) number of object queries, and (3) loss weight ratios among class and mask prediction losses. Results are summarized in Tab. 5, Tab. 6, and Tab. 7, respectively.

**Learning Rate Ratio.** We find that using the same learning rate for both the CNN encoder and Transformer components leads to unstable training and collapse, as shown in Tab. 5. Specifically, setting the Transformer learning rate equal to the CNN's (1:1) results in model failure. This aligns with prior findings that Transformers often require lower learning rates in low-data regimes. The best performance (DSC = 0.913) is achieved when the Transformer learning rate is set to 1/10 of the CNN's (*i.e.,* 1:0.1), while further reduction to 1:0.05 slightly underperforms at 0.908 DSC.

**Number of Object Queries.** In contrast to prior works like MaskFormer and Mask2Former on natural images, where object queries are often over-provisioned to ensure full coverage, we observe

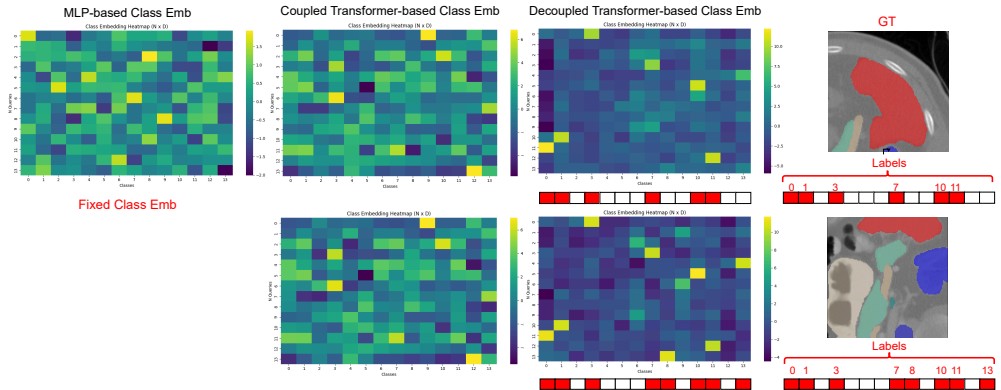

Figure 3: Visualization of Class Embedding for different segmentation heads.

that in medical image segmentation, using a minimal number of object queries, which is equal to the number of semantic classes, yields the best performance and fastest convergence. In Tab. 6, using exactly $N$ queries gives the highest DSC of 0.913, while increasing the number to $2N$, $3N$, or $4N$ leads to consistent degradation in performance (DSC drops to 0.904, 0.900, and 0.891, respectively).

**Loss Ratio Sensitivity.** We ablate the relative weights between classification loss ($\lambda_{\text{CLS}}$), binary cross-entropy for masks ($\lambda_{\text{Mask}_{\text{BCE}}}$), and Dice loss ($\lambda_{\text{Mask}_{\text{Dice}}}$). As shown in Tab. 7, the optimal configuration is found to be 2:10:10, achieving a DSC of 0.913. Using lower mask loss weights (e.g., 5:5) results in under-optimized segmentation masks (DSC 0.902), while increasing the classification weight to 4 degrades class discriminability (DSC 0.889 and 0.910, respectively for 4:5:5 and 4:10:10).

### 4.4 VISUALIZATION OF CLASS EMBEDDINGS

We present the visualizations of class embeddings under two different input images in the first and second rows of Figures 3, respectively. The rightmost column shows the ground truth (GT) for reference. We display only a single slice for clarity, along with the corresponding image labels. Since the MLP-based class become fixed after training, we only present their visualizations once. In these visualizations, the x-axis in Figure 3 represents the class indices (for class embeddings). The y-axis denotes the number of queries. For MLP-based class embeddings, once trained, the vectors tend to become fixed and nearly averaged across channels and each query shows moderate activations across all dimensions. In contrast, our data-driven Transformer-based class embeddings are dynamically adapted to the input data distribution. More importantly, in the decoupled formulation, the learned class embeddings demonstrate explicitly semantic separation: specific labels exhibit stronger activations along distinct channels, rather than relying on fixed channel indices to represent semantic classes. As shown in the Figure 3, the prominent yellow activation blocks are sharply aligned with specific classes, indicating that the decoupled class embeddings capture distinct semantic concepts with clear channel-wise activation patterns. This contrasts with the MLP-based baseline, where the activations are more diffuse and less semantically structured.

## 5 CONCLUSION AND FUTURE WORK

In conclusion, we present MaskMed, a new paradigm that breaks away from traditional segmentation head designs by decoupling mask generation and class prediction. Our unified architecture, enhanced with a full-scale aware deformable transformer for effective fusion of hierarchical encoder features, significantly improves segmentation accuracy and robustness. MaskMed achieves state-of-the-art performance on AMOS 2022 (91.3% Dice) and BTCV (87.2%). We believe MaskMed opens new directions for building more flexible and interpretable medical segmentation models.

**Future Work.** We believe that increasing the number of object queries could further improve performance by allowing richer and more redundant representations, though this requires addressing potential convergence issues. A larger query pool also enables label augmentation, where each class is augmented with multiple mask variants to guide different queries. While our initial attempt to apply contrastive loss between mask and class embeddings was unsuccessful, it remains a promising direction. Moreover, our model shows strong potential in handling multi-instance segmentation for classes with complex anatomical variability, such as brain neuron segmentation.

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

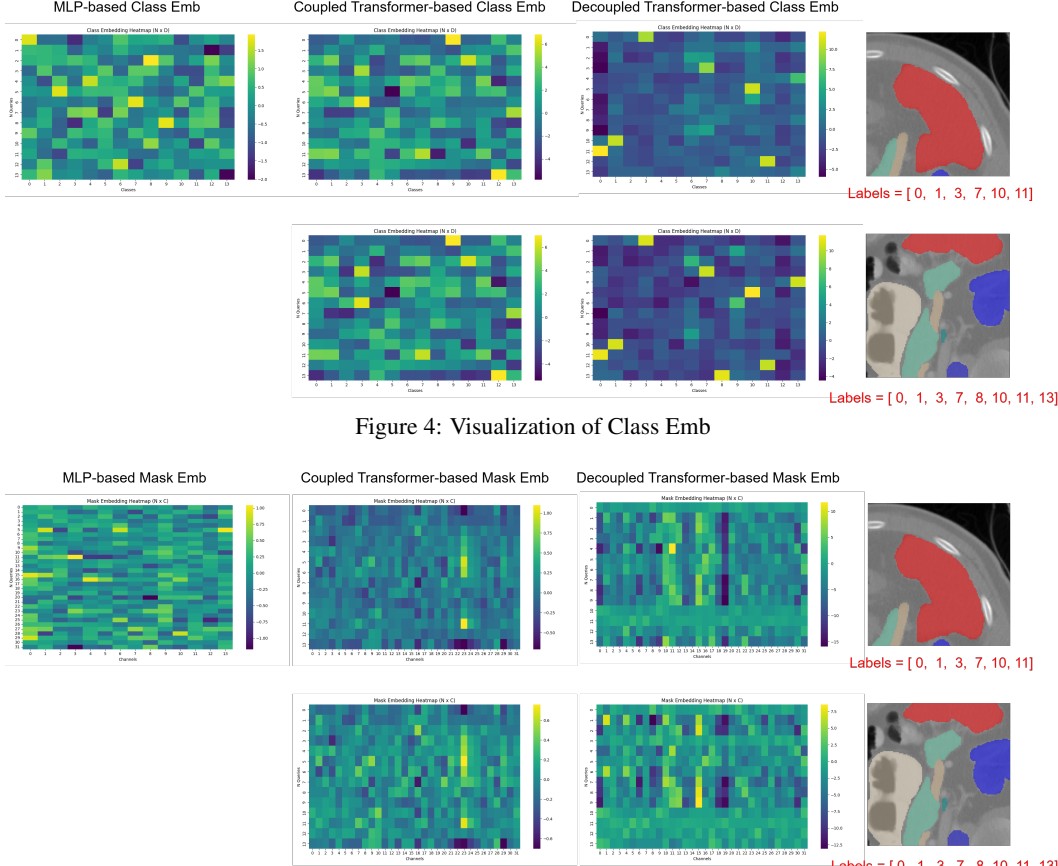

Figure 4: Visualization of Class Emb

Figure 5: Visualization of Mask Emb

# A   VISUALIZATION FOR MASK EMB AND CLASS EMB

We present the visualizations of mask embeddings and class embeddings under two different input images in the first and second rows of Figures 4, 6, respectively. The rightmost column shows the ground truth (GT) for reference. We display only a single slice for clarity, along with the corresponding image labels. Since the MLP-based class and mask embeddings become fixed after training, we only present their visualizations once. In these visualizations, the x-axis in Figure 4 and Figure 6 represents the class indices (for class embeddings) and feature channels (for mask embeddings), respectively. The y-axis denotes the number of queries.

## A.1   VISUALIZATION OF CLASS EMBEDDINGS

We visualize the learned class embeddings in Figure 4. For MLP-based class embeddings, once trained, the vectors tend to become fixed and nearly averaged across channels and each query shows moderate activations across all dimensions. In contrast, our data-driven Transformer-based class embeddings are dynamically adapted to the input data distribution. More importantly, in the decoupled formulation, the learned class embeddings demonstrate clear semantic separation: specific labels exhibit stronger activations along distinct channels, rather than relying on fixed channel indices to represent semantic classes. As shown in the Figure 4, the prominent yellow activation blocks are sharply aligned with specific classes, indicating that the decoupled class embeddings capture distinct semantic concepts with clear channel-wise activation patterns. This contrasts with the MLP-based baseline, where the activations are more diffuse and less semantically structured.

## A.2   VISUALIZATION OF MASK EMBEDDINGS

We visualize the learned class embeddings in Figure 6. Our decoupled Data-driven Transformer-based mask embeddings exhibit diverse patterns across the channel dimension, reflecting spatially adaptive

Figure 6: Details of our FSAD-Transformer.

and query-specific attention. This contrasts with point-wise convolution heads, which rely on fixed filters and lack flexibility.

## B DETAILS OF DEFORMABLE TRANSFORMER

In Figure 6, we present the details of our Full-Scales Awared Transformer (FSAD-Transformer). Full-scale features are first passed through the proposed FSAD-Attention module, followed by a LayerNorm and a residual connection. This is then fed into a Feed-Forward Network (FFN) and another LayerNorm, which is again followed by a residual connection. The overall structure resembles a standard Transformer block, but with the self-attention mechanism replaced by our proposed FSAD-Attention.

**FSAD-Attention.** The full-scale features are used as the value, while the smallest three scales among them are selected as the key. The key features are passed through two separate MLP layers to generate the attention weights and offsets, respectively. The value features are processed by an MLP to produce an output, which is then sampled using the learned offsets via a grid sampling operation. The sampled features are subsequently weighted by the attention weights to produce the final output.

## C IMPLEMENTATION DETAILS

We adopt the nnUNet framework for training, modifying only the network architecture while keeping all other configurations consistent. Data augmentation strategies follow those used in nnUNet. The initial learning rate is set to 0.001, and we apply a polynomial decay strategy as defined in Eq. equation 7:

$$lr_i(e) = \lambda_i \cdot init\_lr \cdot \left(1 - \frac{e}{\text{MAX\_EPOCH}}\right)^{0.9}, \quad i \in \{\text{CNN}, \text{Transformer}\} \tag{7}$$

where $e$ denotes the current epoch, and MAX_EPOCH is set to 1000, with each epoch consisting of 250 iterations. And $\lambda_i$ is a scaling factor that controls the learning rate ratio between the CNN and Transformer modules. We set $\lambda_{\text{CNN}} = 1.0$ and $\lambda_{\text{Trans}} = 0.1$ in our experiments. We use SGD as the optimizer with a momentum of 0.99. The weight decay is set to $3 \times 10^{-5}$. The batch size is set to 2 for all experiments. The loss function is a combination of cross-entropy loss and Dice loss. We employ a 5-fold cross-validation strategy for results presented in Tab. 1.

## D ANALYSIS OF OUR MASKMED

### D.1 COMPARISONS WITH MASKFORMER AND MASK2FORMER

In this section, we compared our MaskMed with MaskFormer and Mask2Former. Figure 7 illustrates the comparison details. The main distinctions of our method from these baselines are as follows:

- **UNet as Backbone**: Directly training a 3D version of Mask2Former leads to unstable optimization and frequent collapse, due to the mismatch between CNN encoders and transformer decoders. Meanwhile, the two models utilizes the Feature Pyramid Network (FPN) to gradually upscale the image features, which is different from traditional UNet architecture. Therefore, we build on the widely adopted UNet in medical image segmentation as the backbone and introduce

Figure 7: Compared our MaskMed with MaskFormer and Mask2Former.

segmentation heads (SegHead) to decouple class and mask embeddings. To stabilize training, we apply differentiated learning rates for CNN-based modules (UNet) and Transformer-based modules (SegHead and FSAD-Transformer).

- **Multi-scale interaction between Class and Mask Embeddings**: In MaskFormer and Mask2Former, mask predictions are produced only at the final resolution, where the highest-resolution FPN features are multiplied by mask embeddings from different layers in the transformer decoder. Both mask and class embeddings are derived from interactions between the low-resolution FPN input and an initial query, leading to a substantial semantic gap between them. In contrast, our method introduces stage-wise interactions between shared class embeddings and multi-scale decoder features from the UNet, significantly shortening the semantic path and improving alignment between class and mask representations.

- **FSAD-Transformer for full-scale feature aggregation**: We propose the FSAD-Transformer, a novel module that aggregates skip connections from all encoder stages into a unified Transformer block. This design enables global feature fusion across scales, which is especially effective in capturing long-range dependencies. In contrast, Mask2Former employs a Deformable Transformer that leverages only the final three encoder features.

- **3D MaskMed vs. 2D MaskFormer and Mask2Former**: All modules in our framework are implemented in 3D, making them more suitable for volumetric medical image segmentation compared to the 2D designs of MaskFormer and Mask2Former.

Our framework is inspired by Mask2Former and DETR. However, to the best of our knowledge, no prior work in the past four years has successfully applied a decoupled class/mask prediction paradigm to 3D medical image segmentation, let alone demonstrated strong performance. Our work establishes this first successful adaptation, supported by both technical innovations and empirical results.

We believe our work opens up a new research direction by successfully demonstrating the feasibility of a decoupled paradigm in 3D medical image segmentation. While our method addresses several key challenges, it also reveals many open questions worth further investigation. We hope our work lays a solid foundation for future research in this direction.

## D.2 THE NECESSITY OF OUR DECOUPLED FRAMEWORK

We believe it is necessary for the medical image segmentation community to consider adopting this design for the following key reasons:

- **Flexibility**: Our decoupled framework allows separate modeling of class and mask predictions, enabling task-specific customization. Unlike traditional architectures that predict all categories simultaneously via a softmax function, our method can easily adapt to different segmentation scenarios.

- **Interpretability**: As shown in our supplementary material, the class embeddings capture the semantic information explicitly, and the mask embeddings explicitly capture spatial mask structures. The decouple method provides valuable interpretability, offering a new paradigm for understanding medical segmentation models.

- **Strong Performance**: Our decoupled design, particularly the addition of SegHead and FSAD-Transformer, consistently achieves state-of-the-art results across multiple benchmarks. This demonstrates that the proposed framework is not only theoretically motivated but also practically effective.

In summary, our approach introduces a flexible, interpretable, and high-performing segmentation paradigm that we believe can benefit future research in medical image segmentation.

# E   USAGE OF LARGE LANGUAGE MODELS

In this paper, we only use Large Language Models to aid or polish writing.

