# OpenReview forum: "MaskMed: Decoupled Mask and Class Prediction for Volumetric Medical Image Segmentation"
_ICLR.cc/2026/Conference — ICLR 2026 Conference Withdrawn Submission_

### Official Review · Reviewer_bFpU · 2025-10-29

**Soundness:** 2
**Presentation:** 2
**Contribution:** 2
**Rating:** 2
**Confidence:** 5

**Summary:**

(1) Propose MaskMed, a unified segmentation head framework that generalizes traditional point-wise segmentation heads into a fully decoupled design, where binary mask prediction and class prediction are independently learned and jointly optimized
(2) Introduce bipartite matching between predicted mask-label pairs and ground truth, marking the first application of this strategy in medical image segmentation to establish precise supervision
(3) Develop a Masked Multi-Scale Segmentation Head that uses a shared query mechanism across decoder stages, enabling consistent semantic prediction while refining spatial details hierarchically with masked attention
(4) Introduce the Full-Scale Aware Deformable Transformer to efficiently fuse full-scale encoder features by attending to sparse yet informative regions across all spatial resolutions

**Strengths:**

(1) The methods and architectures are well-described to help readers better understand their methods
(2) Authors tried to compare their methods with other existing methods to demonstrate the differences of their methods.
(3) Ablation study results are fully described.

**Weaknesses:**

(1) The overall improvement in medical image segmentation is limited. The proposed method demonstrated a close segmentation accuracy with other existing methods. These baseline models were mainly proposed before 2023. Thus, If the proposed methods are compared with other segmentation models which were proposed from 2023 to 2025, it will not demonstrate superior performance. Since it demonstrated a similar performance, it is necessary to provide standard deviations and statistical analysis results. Medical image segmentation is a very popular research field, so more SOTA methods proposed from 2023 to 2025 need to be compared.
(2) The overall theoretical contribution is low. Authors proposed a fully-scaled aware deformable transformer by designing modules and incorporating them into a transformer architecture for medical image segmentation. This modification from N to N+1 does not make fundamental changes or tackle current challenges in a fundamental ways.
(3) Authors only evaluated their methods on abdominal CT images, but they claimed that their methods were proposed for medical image segmentation. Thus, it is necessary to evaluate their methods on other imaging modalities or tune down their statement.
(4) Qualitative comparison or visualization of segmentation results are missing.
(5) Computational complexity is not provided.

**Questions:**

(1) The overall improvement in medical image segmentation is limited. The proposed method demonstrated a close segmentation accuracy with other existing methods. These baseline models were mainly proposed before 2023. Thus, If the proposed methods are compared with other segmentation models which were proposed from 2023 to 2025, it will not demonstrate superior performance. Since it demonstrated a similar performance, it is necessary to provide standard deviations and statistical analysis results. Medical image segmentation is a very popular research field, so more SOTA methods proposed from 2023 to 2025 need to be compared.
(2) The overall theoretical contribution is low. Authors proposed a fully-scaled aware deformable transformer by designing modules and incorporating them into a transformer architecture for medical image segmentation. However, the overall segmentation pipeline was proposed by nnUNet, and this modification from N to N+1 does not make fundamental changes or tackle current challenges in a fundamental ways.
(3) Authors only evaluated their methods on abdominal CT images, but they claimed that their methods were proposed for medical image segmentation. Thus, it is necessary to evaluate their methods on other imaging modalities or tune down their statement.
(4) Qualitative comparison or visualization of segmentation results are missing.
(5) Computational complexity is not provided.

---

### Official Review · Reviewer_oacz · 2025-10-30

**Soundness:** 2
**Presentation:** 2
**Contribution:** 1
**Rating:** 2
**Confidence:** 5

**Summary:**

In this study, the authors focused on medical image segmentation models that used a point-wise conv head. The method can provide a hard-coded codded to one channel to class mapping during segmentation. The authors highlight the class agnostic mask prediction and semantic class prediction, which can learn independently and jointly optimized. The model has a decoupled segmentation head, which uses a single shared query set across all decoder stages.

**Strengths:**

- The decouped segmentation head is adopted for 3D medical image segmentation tasks.
- A nice try of transformer-based model for medical image segmentation, as CNN was still dominant in the domain for many tasks.
- The authors have conducted comprehensive ablation studies and analysis.
- The use of a mask classification paradigm is new in the medical image analysis research.

**Weaknesses:**

- Limited insight on medical image segmentation, the variant of transformer variant and adoption of a natural image segmentation method to - 3D medical image is relatively constrained.
Though useful in general vision tasks, the mask classification is not the first time introduced in medical segmentation; DETR-style methods are well adopted in earlier studies.
- Query-based and prompt-based studies are emerging in recent year, this work lacks a broader literature review and comparisons, such as language-based VLMs.
- The 3D adaptation of a maskFormer is not very attractive and innovative in implementation.

**Questions:**

Questions and suggestions are associated with the weakness section. Thanks.

---

### Official Review · Reviewer_wH3v · 2025-10-31

**Soundness:** 3
**Presentation:** 3
**Contribution:** 2
**Rating:** 2
**Confidence:** 5

**Summary:**

The paper proposes MaskMed, a 3D medical image segmentation framework that separates mask prediction from class assignment within a UNet-style architecture. It introduces a segmentation head that reuses a single set of queries across decoder stages, refining predictions hierarchically and using lower-resolution masks as spatial priors for higher-resolution attention. To improve feature fusion, the method employs a Full-Scale Aware Deformable Transformer, which aggregates information from all encoder scales through deformable attention for efficient global context modeling. The training strategy applies bipartite matching once at the final stage and reuses assignments for deep supervision, with adjustments such as differentiated learning rates and query count tailored to medical imaging. Experiments on AMOS22 and BTCV show consistent improvements over strong baselines, indicating that multi-scale query propagation and full-scale deformable fusion are effective for volumetric segmentation.

**Strengths:**

1. The method was compared against several leading baselines, including nnUNet, SwinUNETR, MaskFormer, and Mask2Former.

2. Ablation studies were conducted on the architecture, optimization sensitivity, and hyperparameters.

3. Visualizations of class and mask embeddings demonstrate clear semantic separation and spatial adaptability, supporting the claim of enhanced interpretability.

**Weaknesses:**

1. The decoupled mask–class paradigm is inspired by MaskFormer/Mask2Former. Even in the medical field, similar ideas have already been explored in \url{https://arxiv.org/pdf/2405.16328?}. The novel contributions of this paper may lie in the segmentation head---specifically, progressive refinement---and feature fusion. However, these areas have been widely explored and are secondary in my opinion.

2. The reported baseline performances appear lower than expected. The proposed method’s performance, in many cases, is comparable to these baseline numbers reported in multi-organ segmentation studies. Notably, some papers (e.g., the above one) achieve similar performance using weakly supervised learning, unlike the fully supervised training employed here.

3. FSAD-Transformer builds on deformable attention from DETR, with its novelty mainly in integration and application.

4. While the architectural ablations are thorough, the paper does not explore alternative query initialization strategies or contrastive learning approaches, which are mentioned as future work.

5. Some figures (e.g., Figures 1 and 2) are dense and difficult to interpret without zooming in.

**Questions:**

1. Beyond operating in 3D and within a UNet, could you clarify the minimal set of mechanisms or original contributions that are indispensable? What breaks if any of these components are removed?

2. What is the number of sampling points K per query per scale?

3. How are low-resolution masks converted to attention masks at the next stage? Please give the details.

4. You mention collapse when using the same learning rate for CNN and Transformer blocks. Did you experiment with warm-up, gradient clipping, cosine decay, or AdamW for the Transformer? How often does training collapse with your final recipe?

5. Did you re-run baselines using your learning rate ratio, deep supervision, and matching-once strategy to ensure a fair comparison? Some baseline methods' performances are lower than I expected.

6. When training on AMOS22 and evaluating zero-shot on BTCV (and vice versa) without fine-tuning, how does the method compare to nnUNet or Mask2Former under the same protocol?

7. You observe that N queries work best. Could you show results when some classes are missing (partial label sets) and when 2N queries are allowed but duplicates are suppressed via matching?

8. Since the current formulation predicts N masks for N classes, how could it be extended to handle unknown instance counts (e.g., lesions) without over-provisioning queries?

---

### Official Review · Reviewer_2kHF · 2025-11-01

**Soundness:** 2
**Presentation:** 3
**Contribution:** 2
**Rating:** 4
**Confidence:** 4

**Summary:**

The paper introduces MaskMed, a novel architecture for 3D medical image segmentation that rethinks the traditional segmentation head design in U-Net–like models. Instead of using the standard point-wise convolutional head (where each output channel is tied to a specific class), the authors propose a decoupled segmentation framework that separates mask prediction (class-agnostic binary masks) from class prediction (semantic labels). This approach allows the network to flexibly associate spatial masks with semantic categories, inspired by the Mask2Former / DETR paradigm but extended to the 3D volumetric medical domain.

**Strengths:**

1. Clear motivation and architectural reasoning
- The authors correctly identify a long-standing limitation in segmentation heads, rigid channel-to-class mappings that prevent feature sharing and contextual reasoning.The move to a decoupled mask/class formulation is well motivated and consistent with trends in vision segmentation (Mask2Former, DETR).
2. First adaptation of decoupled mask–class prediction to 3D medical imaging
- Prior decoupled models (MaskFormer, Mask2Former) were designed for 2D natural images. The paper convincingly demonstrates a 3D adaptation that remains stable and performant under limited data conditions.

3. Technically sound and well-engineered solution
- The FSAD-Transformer design elegantly integrates multi-scale encoder features through deformable attention, reducing quadratic cost.
- The shared query mechanism across decoder stages enforces semantic consistency and hierarchical refinement.

**Weaknesses:**

1. Limited conceptual novelty beyond adaptation
- The proposed method largely repackages Mask2Former / DETR-style decoupled heads and deformable attention for the 3D medical domain.
- While the engineering execution is solid, the conceptual leap is incremental, MaskMed is closer to a robust adaptation than a fundamentally new paradigm.
2. Engineering-driven rather than principle-driven
- The paper relies on many interacting modules (shared queries, deformable transformers, masked attention). It’s unclear which components contribute fundamentally to generalization versus adding optimization flexibility.

**Questions:**

1. On experimental generalization
- Have the authors evaluated the masked multi-scale segmentation head on other network backbones, particularly nnU-Net, which also computes losses at multiple decoder scales?
- It would be helpful to understand whether the proposed decoupled head consistently improves performance across architectures, or if its benefit is coupled to the specific backbone used in MaskMed.
2. On query design
- Why does using exactly 1×N queries (equal to the number of classes) yield the best results? Is this choice purely heuristic, or is there a theoretical rationale suggesting minimal query redundancy is optimal for medical segmentation tasks?
3. On baseline comparisons
- Since the proposed architecture leverages deformable attention, it would strengthen the experimental validation to include a 3D deformable convolutional network baseline (e.g., DeformUX-Net) for fair comparison on similar medical segmentation tasks. This would clarify whether the improvements stem from the deformable transformer itself or from the decoupled segmentation head design.
4. On novelty and core contribution
- Beyond the engineering insights in modifying the segmentation head, can the authors more clearly articulate the core conceptual innovation of this work?
- As the method builds upon previously proposed networks, additional discussion on its distinct impact or theoretical advancement for the medical image segmentation domain would help establish its novelty and significance.

---

### Note · Authors · 2025-11-14

I have read and agree with the venue's withdrawal policy on behalf of myself and my co-authors.